# Diversity of Alien Macroinvertebrate Species in Serbian Waters

**Katarina Zorić *** , **Ana Atanacković, Jelena Tomović, Božica Vasiljević, Bojana Tubić and Momir Paunović**

Department for Hydroecology and Water Protection, Institute for Biological Research "Siniša Stanković"—National Institute of Republic of Serbia, University of Belgrade, Bulevar despota Stefana 142, 11060 Belgrade, Serbia; adjordjevic@ibiss.bg.ac.rs (A.A.); jelena.tomovic@ibiss.bg.ac.rs (J.T.); bozica@ibiss.bg.ac.rs (B.V.); bojana@ibiss.bg.ac.rs (B.T.); mpaunovi@ibiss.bg.ac.rs (M.P.)
* Correspondence: katarinas@ibiss.bg.ac.rs

**Abstract:** This article provides the first comprehensive list of alien macroinvertebrate species registered and/or established in aquatic ecosystems in Serbia as a potential threat to native biodiversity. The list comprised field investigations, articles, grey literature, and unpublished data. Twenty-nine species of macroinvertebrates have been recorded since 1942, with a domination of the Ponto-Caspian faunistic elements. The majority of recorded species have broad distribution and are naturalized in the waters of Serbia, while occasional or single findings of seven taxa indicate that these species have failed to form populations. Presented results clearly show that the Danube is the main corridor for the introduction and spread of non-native species into Serbia.

**Keywords:** Serbia; inland waters; allochthonous species; introduction

## 1. Introduction

The Water Framework Directive (WFD) [1] represents key regulation and one of the most important documents in the European Union water legislation since it was adopted in 2000. A primary purpose of the directive was to improve and integrate the way water bodies are managed across Europe with the principal aim of achieving at least "good ecological and chemical status" for all water bodies. The original target for achieving good status was 2015, but it has been extended to 2021 or 2027 [2].

The ecological status of inland waters is one of the two main determinants of the WFD [1]. Aquatic macroinvertebrates are a very important component of freshwater ecosystems and one of the five biological quality elements. Although the text of the directive does not specifically mention alien or invasive species, it is obvious that they could pose great pressure and act as a threat to aquatic communities and ecosystems and indirectly affect the achievement of good status of water bodies [3]. Serbia, as a candidate country for entry to the European Union since 2012, must implement all EU directives, including the WFD with its specific goals in water protection, improvement, and sustainable use.

Over the past 20 years, research on allochthonous macroinvertebrate species has intensified in Serbia. A high level of biocontamination in aquatic ecosystems has already been confirmed and allochthonous species have been recorded among invertebrates and vertebrates [4]. Previous research on allochthonous macroinvertebrate species was focused on individual species. A detailed overview of only half of allochthonous species is given up to now: *Branchiura sowerbyi* [5,6], *Corbicula fluminea* and *C. fluminalis* [7], *Craspedacusta sowerbii* [8], *Dreissena rostriformis bugensis* [9], *Faxonius limosus* [10], *Hypania invalida* [11], *Eriocheir sinensis* [12,13], *Manayunkia caspica* [14], *Physa acuta* [15–17] and

*Sinanodonta woodiana* [18]. In addition, allochthonous species of different aquatic systems in Serbia were the subject of several publications, and the most comprehensive research was carried out on the Danube, Sava, Tisa, and Velika Morava rivers [4,19–27].

Based on the obtained data, the following goals have been set—identification and review of the present allochthonous macroinvertebrate species—providing the list of alien taxa, taxonomic and zoogeographic analyses of allochthonous species, and possible ecological implications.

## 2. Materials and Methods

In Serbia, there are thousands of watercourses with a total length of 65,980 km [28] predominantly small and medium-sized rivers, up to 100 km in length. According to [29] four groups of watercourses have been differentiated. The first group is typical lowland, slow-flowing rivers—the Danube, Sava, and Tisa. The second group is comprised of rivers within the Velika Morava basin, and the third group is represented by the Drina, Lim, and Nera rivers. The Nišava River was found to be a separate system.

Limnological investigations of Serbian waters were initiated in 1947 and have continued up to now. The present paper brings together all of the available information on the allochthonous macroinvertebrate species in the watercourses of Serbia. The list was compiled from peer-reviewed articles, monographs, books, and grey literature (reports, thesis, and unpublished data). Further, comprehensive field investigations were performed in the period 2000–2019 at a total of 402 sampling sites.

The study of aquatic ecosystems covered the entire length of all major river basins. The river system of Serbia includes ten main drainages, while detailed research has been performed on a total of nine of them—the Danube, Sava, Drina, Kolubara, Zapadna Morava, Južna Morava, Velika Morava, Timok and rivers belonging to the Aegean basin. Although the Tisa river is a tributary of the Danube, it has been presented as a separate river system regarding numerous records of allochthonous species. The most intense research was conducted in the Danube, Sava, Tisa, and Velika Morava basins. Since 2000, three international surveys [30–32], numerous national investigations, and routine monitoring surveys were carried out that resulted in the most detailed and precise data on alien fauna. However, during the last ten years, research of small and medium-sized watercourses in the rest of the region has also intensified, encompassing the entire territory of Serbia and providing an up-to-date inventory of alien species. During field investigations, benthic samples were collected with a benthic hand net (mesh size 500 μm), benthic dredge (mesh size 250 μm), and Van Veen grab (270 cm$^2$). Additionally, free diving was carried out at depths between 0.4 and 7 m to obtain specimens of malacofauna. The sampling method was conditioned by the type of examined waterbody. Sampling in shallow bank region with benthic hand net was conducted according to multihabitat sampling procedure [33] from all available types of microhabitat representing with more than 5% of the total habitat area along the sampling stretch (100 m). Grab and dredge were used only in cases where samples could not be collected by hand net (great depth or flow velocity). In the case of samples collected by Van Veen grab, the animals were separated from the sediment using a 200-μm sieve.

The list of allochthonous species was compiled on the basis of the currently valid taxonomic nomenclature using the World Registered of Marine Species [34]. The classification of allochthonous species included zoogeographic and horologic analyses of species, as well as analysis of the pathway of introduction. The year of the first discovery of an introduced species was based on the date of the first publication.

## 3. Results

In watercourses of Serbia, a total of 29 allochthonous species of macroinvertebrates were recorded (Table 3, Figure 1).

**Table 1.** Allochthonous macroinvertebrate species in watercourses of Serbia. (JDS 3—Joint Danube Survey 3).

| Class, Family and Species Name | Native Area | Year of First Record | Location (River, Site Name, Kilometer) | Coordinates | Reference |
|---|---|---|---|---|---|
| Bivalvia Corbiculidae 1. *Corbicula fluminalis* (O. F. Müller, 1774) | Asia | 1998 | Danube, Orešac, 1126 | N 44°39′41.0″ E 20°48′05.0″ | [12] |
| 2. *Corbicula fluminea* (O.F. Müller, 1774) | Asia | 2001 | Danube, Stara Palanka, 1077 | N 44°49′17.0″ E 21°20′00.0″ | [12] |
| Dreissenidae 3. *Dreissena rostriformis bugensis* (Andrusov, 1897) | Ponto-Caspian region | 2010 | Danube, Veliko Gradište, 1059 | N 44°46′16.4″ E 21°31′20.8″ | [9] |
| 4. *Dreissena polymorpha* (Pallas, 1771) | Ponto-Caspian region | 1986 | Danube, Novi Sad, 1250 | N 45°12′37.6″ E 19°56′18.8″ | [35] |
| Unionidae 5. *Sinanodonta woodiana* (Lea, 1834) | Asia | 1998 | Danube armlet, 1159 | N 44°50′22.2″ E 20°35′20.0″ | [18] |
| Gastropoda Physidae 6. *Physella acuta* (Draparnaud, 1805) | North America | 1986 | Danube, Tekija, 956 | N 44°41′19.2″ E 22°24′56.8″ | [35] |
| Planorbidae 7. *Ferrissia fragilis* (Tryon, 1863) | North America | 2010 | Danube, Dubovački rit, 1087 | N 44°47′07.3″ E 21°15′44.6″ | [16] |
| Malacostraca Cambaridae 8. *Faxonius limosus* (Rafinesque, 1817) | North America | 2004 | Danube, Smederevo, 1112 | N 44°41′31.6″ E 20°57′38.5″ | [10] |
| Varunidae 9. *Eriocheir sinensis* H. Milne-Edwards, 1853 | Asia | 1973 | Tisa, Novi Bečej, 72 | N 45°35′36.0″ E 20°07′56.0″ | [13] |
| Corophiidae 10. *Chelicorophium curvispinum* (G. O. Sars, 1895) | Ponto-Caspian region | 1942 | Danube, Smederevo, 1116 | N 44°41′57.9″ E 20°58′10.1″ | [36] |

**Table 2.** Allochthonous macroinvertebrate species in watercourses of Serbia. (JDS 3—Joint Danube Survey 3).

| Class, Family and Species Name | Native Area | Year of First Record | Location (River, Site Name, Kilometer) | Coordinates | Reference |
|---|---|---|---|---|---|
| 11. *Chelicorophium robustum* (G. O. Sars, 1895) | Ponto-Caspian region | 2007 | Danube, downstream Pančevo, 1151 | N 44°49′31.4″ E 20°39′01.4″ | [37] |
| Gammaridae 13. *Dikerogammarus bispinosus* Martynov, 1925 | Ponto-Caspian region | 2001 | Danube, Bačka Palanka, 1300 | N 45°14′02.4″ E 19°23′05.5″ | [29] |
| 14. *Dikerogammarus haemobaphes* (Eichwald, 1841) | Ponto-Caspian region | 2001 | Danube, Bezdan, 1429 | N 45°52′06.6″ E 18°50′15.1″ | [20] |
| 15. *Dikerogammarus villosus* (Sowinsky, 1894) | Ponto-Caspian region | 2001 | Danube, Bezdan, 1429 | N 45°52′06.6″ E 18°50′15.1″ | [20] |
| 16. *Echinogammarus ischnus* (Stebbing, 1899) | Ponto-Caspian region | 2007 | Danube, Golubac, 1040 | N 44°39′30.3″ E 21°40′18.4″ | [4] |
| Janiridae 17. *Jaera sarsi* Valkanov, 1936 | Ponto-Caspian region | 2001 | Danube, Bezdan, 1429 | N 45°52′06.6″ E 18°50′15.1″ | [20] |
| Mysidae 18. *Hemimysis anomala* G. O. Sars, 1907 | Ponto-Caspian region | 2005 | Danube, Veliko Gradište, 1059 | N 44°46′01.6″ E 21°30′53.6″ | [38] |
| 19. *Katamysis warpachowskyi* G. O. Sars, 1893 | Ponto-Caspian region | 2005 | Danube, Veliko Golubinje, 986 | N 44°28′00.3″ E 22°10′21.6″ | [38] |
| 20. *Limnomysis benedeni* (Czerniavsky, 1882) | Ponto-Caspian region | 1977 | Tisa, near Titel, 9 | N 45°12′19.8″ E 20°18′47.2″ | [38] |
| 21. *Paramysis lacustris* (Czerniavsky, 1882) | Ponto-Caspian region | 1999–2001 | Danube, upstream from Iron gate 1083–1071 | N 44°47′07.3″ E 21°15′44.6″– N 44°48′57.0″ E 21°21′40.7″ | [39] |

**Table 3.** Allochthonous macroinvertebrate species in watercourses of Serbia. (JDS 3—Joint Danube Survey 3).

| Class, Family and Species Name | Native Area | Year of First Record | Location (River, Site Name, Kilometer) | Coordinates | Reference |
|---|---|---|---|---|---|
| Pontogammaridae 22. *Obesogammarus obesus* (G. O. Sars, 1894) | Ponto-Caspian region | 2001 | Danube, Bezdan, 1429 | N 45°52′06.59″ E 18°50′15.10″ | [20] |
| Oligochaeta Tubificidae 23. *Branchiura sowerbyi* Beddard, 1892 | West Pacific | 1972 | Fish pond near Futog, 1270 | N 45°14′20.4″ E 19°40′08.6″ | [5] |
| 24. *Potamotrix moldaviensis* Vejdovský and Mrázek, 1903 | Ponto-Caspian region | 2007 | Danube, near Dalj, 1355 | N 45°29′50.2″ E 19°00′35.8″ | [40] |
| Polychaeta Ampharetidae 25. *Hypania invalida* (Grube, 1860) | Ponto-Caspian region | 1971 | Sava, Makiš, 10 | N 44°46′21.0″ E 20°21′05.0″ | [11] |
| Fabriciidae 26. *Manayunkia caspica* Annenkova, 1929 | Ponto-Caspian region | 1943 | Danube, Tekija, 956 | N 44°41′20.0″ E 22°24′53.0″ | [14] |
| Turbellaria Planariidae 27. *Dugesia tigrina* (Girard, 1850) | North America | 2008 | Kolubara, Ćelije | N 44°21′56.2″ E 20°11′53.2″ | [41] |
| Hydrozoa Olindiasidae 28. *Craspedacusta sowerbii* Lankester, 1880 | Asia | 1958 | Pool near Velika Morava River, Ćuprija | N 43°56′49.9″ E 21°22′40.3″ | [42] |
| Phylactolaemata Pectinatellidae 29. *Pectinatella magnifica* (Leidy, 1851) | North America | 2013 | Danube, Novi Sad, 1252 | N 45°15′31.3″ E 19°53′13.6″ | [26] |

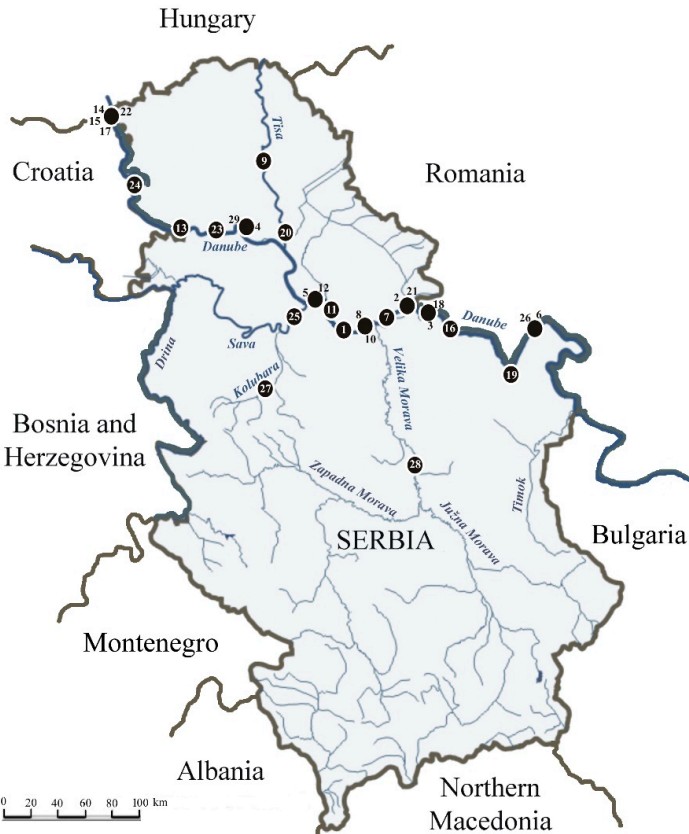

**Figure 1.** Locations of initial findings of allochthonous macroinvertebrate species along the rivers in Serbia. Numbers correspond to the species as listed in the Table 3.

The largest number of species, 18 (62.07%), originates from the Ponto-Caspian region, while Asia and North America are represented by 5 species each (17.24%). One recorded species (3.45%) is from the Western Pacific.

Among 29 recorded species the largest number of allochthonous species belongs to Malacostraca (15 species, Figure 2), while members of Bivalvia are represented with five species. Other six groups (Gastropoda, Oligochaeta, Polychaeta, Turbellaria, Hydrozoa and Phylactolaemata) are represented with a total of nine species.

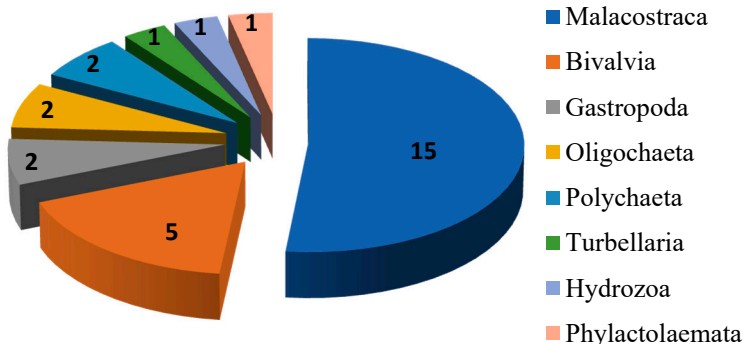

**Figure 2.** Number of alien species per taxonomic group.

Current distribution of the species based on literature data and field investigation in Serbian major river basins is given in Table 4. Distribution of recorded alien species is predominantly observed along the slow-flowing lowland rivers with a dominance of fine sediment (Danube, Sava, Tisa) and Velika Morava. In total, 28 species were recorded in the Danube, followed by Sava (16 species),

Tisa (15 species), and Velika Morava River (10 species). Six macroinvertebrate species were detected in Zapadna Morava basin, while only two alien species were recorded in Južna Morava, Drina, and Kolubara basin, and one species in Timok basin. Spatially, the percentage participation of alien species in the macroinvertebrate community ranged from 0.9% to 33.33% in the Danube, 1.01% to 17.67% in the Sava, 0.33% to 57.58% in the Tisa and 1.03% to 71.12% in the Velika Morava River. In other watercourses, alien species occur irregularly in low numbers.

**Table 4.** Current distribution of allochthonous macroinvertebrate species in Serbian major river basins. Abbreviations: DA–Danube, SA–Sava, DR–Drina, KO–Kolubara, ZM–Zapadna Morava, JM–Južna Morava, VM–Velika Morava, TI–Timok, Tisa–Danube tributary.

| Species | River Basins |
|---|---|
| *Corbicula fluminalis* | DA, SA |
| *Corbicula fluminea* | DA, SA, VM, Tisa |
| *Dreissena rostriformis bugensis* | DA |
| *Dreissena polymorpha* | DA, SA, DR, VM, Tisa |
| *Sinanodonta woodiana* | DA, SA, ZM, VM, Tisa |
| *Physella acuta* | DA, SA, DR, KO, ZM, JM, VM, TI |
| *Ferrissia fragilis* | DA |
| *Faxonius limosus* | DA, SA, VM, Tisa |
| *Eriocheir sinensis* | DA, Tisa |
| *Chelicorophium curvispinum* | DA, SA, ZM, VM, Tisa |
| *Chelicorophium robustum* | DA, SA |
| *Chelicorophium sowinskyi* | DA |
| *Dikerogammarus bispinosus* | DA |
| *Dikerogammarus haemobaphes* | DA, SA, Tisa |
| *Dikerogammarus villosus* | DA, SA, ZM, VM, Tisa |
| *Echinogammarus ischnus* | DA |
| *Jaera sarsi* | DA, SA, VM, Tisa |
| *Hemimysis anomala* | DA |
| *Katamysis warpachowskyi* | DA |
| *Limnomysis benedeni* | DA, SA, ZM, Tisa |
| *Paramysis lacustris* | DA |
| *Obesogammarus obesus* | DA |
| *Branchiura sowerbyi* | DA, SA, KO, ZM, JM, VM, Tisa |
| *Potamotrix moldaviensis* | DA |
| *Hypania invalida* | DA, SA, Tisa |
| *Manayunkia caspica* | DA |
| *Dugesia tigrina* | DA, SA, KO, Tisa |
| *Craspedacusta sowerbii* | SA, VM |
| *Pectinatella magnifica* | DA, Tisa |

The highest number of alien species per sampling site was recorded in the Danube, upstream from the mouth of Velika Morava (14 species). The average number of alien species per river was also the highest in the Danube (8 species) while the average proportion of alien species to a total number of macroinvertebrate taxa was 10.19% in 2007 and much higher in 2013, 25.53%.

The most widespread species in Serbian waters are *Physella acuta* and *Dikerogammarus villosus*. *P. acuta* was recorded at 77 sites in a total of eight river basins, predominantly in small- and medium-sized watercourses up to 500 m above sea level. *D. villosus* was recorded in 48 sites, along the entire course of the Danube, and in localities in the Sava, Tisa, Velika Morava, and a canal network in the northern part of the country. A single finding of two specimens was from Ljubišnica (Zapadna Morava basin). Broad distribution within the covered area was assessed also for *Branchiura sowerbyi*, *Sinanodonta woodiana*, *Corbicula fluminea*, *Dreissena polymorpha Chelicorophiun curvispinum*, and *Faxonius limosus*. Beside distribution in lowland rivers, findings of *D. polymorpha* in the Drina and *C. curvispinum* in the Ljubišnica and Mali Rzav (Zapadna Morava basin) should be pointed out. Occasional or single findings of seven species (*E. sinensis*,

*Craspedacusta sowerbii*, *Dugesia tigrina*, *Hemimysis anomala*, *Katamysis warpacowsky*, *Corbicula fluminalis*, and *Ferrissia fragilis*) indicate that these species have not yet been naturalized in Serbian waters.

## 4. Discussions

The allochthonous species described in this paper provide the first list of alien macroinvertebrate species registered and/or established in water ecosystems in Serbia. In total, 29 species of macroinvertebrates were recorded, which is about 3% of the total aquatic macroinvertebrate fauna of Serbia (995 species) [43] and about 14% of the total number of alien macroinvertebrate species in Europe (201 species) [44]. In the neighboring area, the most detailed overview of aquatic alien macroinvertebrate species was given for Croatia [45,46] with the complete distribution of species along the water bodies, possible pathways of introduction, and level of biocontamination. No other country in the region has complete data on alien aquatic macroinvertebrate species. In Bulgaria, only three invasive mussels [47] were pointed out as species of special concern regarding their rapid expansion of distribution range and invasive potential, while in Romania two alien species of EU concern were detected [48].

The expansion of Ponto-Caspian species was one of the most important biogeographical processes in terms of the biotic homogenization [49,50]. The European network of waterways has enabled the expansion of aquatic species for hundreds of years [51,52], but during the last century, expansion via the canal system (Volga-Don canal and Main-Danube canal) has greatly increased primarily for the species from Ponto-Caspian region, which has led to an increase in the number of allochthonous species from this area in European watercourses.

Most recorded alien species in Serbia have been introduced by natural dispersal through the Danube from neighboring countries. Introduction of three species has been closely connected with aquaculture. The introduction and spread of clams (*Corbicula* sp.) and Chinese pond mussel, *Sinanodonta woodiana*, was connected to the introduction of the Chinese fish complex of species (Grass carp, Prussian carp, Silver carp, and Bighead carp) from China and other Far Eastern countries into Serbian waters and nearby regions in 1970's [18]. Having in mind the large area of occupancy and distribution throughout all main river basins in Serbia of *Physella acuta*, we may assume that its introduction was facilitated by anthropogenic activity and zoohoria (via aquatic birds) as suggested by [16]. Accidental transport with ornamental aquatic plant may be the vector for range expansion of *Craspedacusta sowerbii*.

The presented distribution, numbers of observed alien species in selected watercourses and the localities of the first findings mainly along the Danube clearly indicate that alien species have been expanding from the Danube as the main corridor to other large rivers in Serbia, i.e., the Sava, Tisa, and Velika Morava. For this type of river, the principal components of the community are Gastropoda, Bivalvia, Oligochaeta, Amphipoda, and Hirudinea [29]. Although researches of large rivers were more extensive, the current distribution of alien species, with a larger number of taxa and highest abundance in lowland rivers, is to be expected considering that recorded alien taxa are from groups that prefer this type of habitats and that they were unlikely to be found in hilly-mountainous streams where insects are dominant group [29,43].

As the results of recent studies showed, the Danube and its main tributaries are under the considerable influence of alien species. Along the Danube, the contribution of alien species has steadily increased over time from 10.19% to 25.53%. Having in mind that the Lower Danube could be considered as a native area of distribution of Ponto-Caspian taxa, which are considered as allochthonous in the Middle Danube, average proportion of alien species is much higher in the Serbian part of Middle Danube (upstream of Golubac).

The presence of introduced species of Mollusca was recorded in typical lowland, slow-flowing rivers–the Danube, Sava, and Tisa, as well as the Velika Morava, except for the species *Physella acuta*, which has a widespread distribution on the territory of Serbia [15]. The most frequent were *Sinanodonta woodiana*, *Corbicula fluminea*, *Dreissena polymorpha,* and *P. acuta*, recorded in all four rivers. All seven species of Mollusca were registered in the Danube, five in the Sava and Tisa

(except *Dreissena rostriformis bugensis* and *Ferrissia fragilis*), while only three allochthonous species were recorded in the Velika Morava–*S. woodiana*, *C. fluminea,* and *P. acuta*.

Considering representatives of the genus Corbicula, *C. fluminalis* was recorded in a small number of localities in low abundance. In contrast, *C. fluminea* was present in many localities on the Danube [24], as well as on the Sava [21,29], and along the entire course of the Tisa, and Velika Morava [22,23,25,27,53] as a dominant alien species.

As regards molluscan fauna of the Danube flooding zone at Dubovački Rit, [16] revealed that *D. polymorpha* has the highest frequency of occurrence, followed by *C. fluminea*, *D. rostriformis bugensis*, and *S. woodiana*, while *F. fragilis* and *P. acuta* had very low frequencies of occurrence. This trend in frequency of occurrence was also confirmed during the 2007 and 2013 surveys. During investigations of the Sava River, certain alien taxa were found to have a high abundance (*S. woodiana*) and frequency (*C. fluminea*). Thus, the Asiatic clam, which was found at all sampling locations, was the most important species with regard to frequency of occurrence [21]. In contrast to the Sava River, analyses of the malacofauna of the Tisa until 2011 showed the dominance of native taxa, *Unio pictorum* and *U. tumidus*, while the influence of alien species was not so strong [23]. Research in the Velika Morava River detected the presence of alien Mollusca along the entire watercourse. According to [22] and [25], the most frequent and abundant species were Chinese pond mussel (*S. woodiana*) and Asiatic clam (*C. fluminea*). The spread of *S. woodiana* was also detected in smaller watercourses, reservoirs, and canals throughout Serbia [54]. The negative effects of the introduction of *S. woodiana* are primarily observed in the competition with native species of the Unionidae family [55], not only among adults, but also at larval stages due to preferences for the same host [56]. A similar effect may be expected because species were recorded together with native species: *Anodonta anatina*, *U. pictorum,* and *U. tumidus*.

The allochthonous North American gastropod *Ferrissia fragilis* was found only at the location of Dubovački Rit on the Danube (1087 km) [16]. Besides its presumed primarily anthropogenic introduction [16], some migratory bird species, mainly *Anas platyrhynchos*, *Gallinago gallinago,* and *Gallinula chloropus* [57] have a major impact on the introduction of this species.

Although discovered recently, the rapid spread of *Dreissena rostriformis bugensis* has been reported. By 2011, the most upstream finding of the species for Serbia was at site Ledinci (1260 km) [16], while only two years later, the species was detected in almost all localities of the middle Danube, with the most upstream locality at 1384 km (upstream from the mouth of the Drava River; unpublished data from JDS 3). The number of specimens ranged from 1 to 19. Recent research also illustrated the extensive distribution range of *D.r. bugensis* in the main Danube tributaries, reservoirs, and canals closely connected to the main flow of the river, in almost equal participation in the mussel community with *D. polymorpha*. Negative impacts of the invasive *D. polymorpha* on the unionids population were not found, but according to [58] localized declines in areas where this species has invaded *Anodonta cygnea* habitats are expected. Unlike other alien Mollusca distributed only in lowland rivers, the species *Physella acuta* inhabits various types of aquatic biotopes-rivers, streams, canals, reservoirs, fishponds, and swamps. It is widespread in Serbia, at some localities as dominant Gastropod species [15] which is in concordance with its cosmopolitan distribution [59], probably due to its high tolerance of organic pollution in water and sediment [15].

Among the alien macroinvertebrates, members of the Malacostraca are the most numerous. Of all animal phylum, Arthropoda has the largest percentage of allochthonous species in Europe (23.1%) [44]. Additionally, of all alien macroinvertebrate species in Europe, Arthropoda are the most numerous (104 of 201 species) [44]. Regarding alien Decapoda species in neighboring countries, spiny cheek crayfish (*Faxonius limosus*) was registered in the Hungarian section of the Danube at Budapest [60] and Kopački rit Nature Park [61]. Further colonization of this part of the Danube was occurring both upstream and downstream. Natural dispersion seems to be the most likely path of introduction in the Serbian part of the Danube. According to [10], introduction was probably aided by the ballast water of ships and legal and illegal stocking. The spreading of spiny cheek crayfish may have negative effects on native crayfish populations, as has already been documented for several European

countries [62,63] where *F. limosus* has become predominant. This is primarily because of the fungal plaque (*Aphanomyces astaci*) carried by *F. limosus*, which is lethal to European crayfish [64,65]. In Serbian waters, two native crayfish species are endangered due to the negative effects of the introduction of *F. limosus–Pontastacus leptodactylus* and *Astacus astacus*. Based on the detailed investigation and literature data [66–68] of the distribution of a third native species, the spring crayfish *A. torrentium*, there were no records that *F. limosus* entered mountain streams, the main habitats of this native species. Another crayfish species, *Pacifastacus leniusculus* (the signal crayfish), is at present the most widespread and one of the most successful crayfish invaders in Europe. The presence of the species was documented in Hungary in 2001 [69] and in Croatia along the entire course of the Mura River [70] and in a part of the Korana River [71]. Due to its very rapid rate of spread [72], further expansion is expected in the area, primarily downstream in the Drava River, and most likely in the Danube. Signal crayfish has not been detected in Serbian waters yet, but its presence in nearby countries implies that it can be expected in the near future.

The species *Chelicorophium curvispinum* is today the most widespread Ponto-Caspian species in the Danube [73]. It was detected in the early 1940s [31] near Smederevo and has spread along the entire Serbian stretch of the Danube. During the expedition in 2007, this species was identified at 12 out of 20 examined sites [32], whereas in 2013 findings of the species were more frequent, at 13 out of 16 localities, with a high abundance at almost every locality (unpublished data collected during JDS 3). Natural dispersion at downstream localities and passive expansion via shipping [73] for upstream sections seems to be the most probable route of its spread along the Danube. *C. curvispinum* is today the most widespread Ponto-Caspian amphipod species in Europe [74]. It inhabits different types of aquatic habitats from slow-flowing or stagnant water, on the bottom or on macrophytes, algae, and stones. It has a wide distribution within the river and connected systems, but in the rest of the territory, the species is not detected. Two other alien corophiids *C. robustum* and *C. sowinsky*, are not significant members of the macroinvertebrate community of the Danube, and they are found at only a few localities with low abundance [32], unpublished data collected during JDS 3.

Ponto-Caspian Peracarida (i.e., amphipods, isopods, and mysids) are in general well established in river systems of Serbia, becoming a persistent part of macrozoobenthos community. All three gammarid species were recorded during the JDS 1 in 2001 [20]. *Dikerogammarus villosus* was recorded along the entire course, while two other species, *D. haemobaphes* and *D. bispinosus* had a limited distribution. This distribution pattern was confirmed in later expeditions (unpublished data collected during JDS 2 and JDS 3). After the initial findings in 2001 *D. villosus* became the most dominant amphipod species along the Danube River, and also in the Danube-Tisa-Danube canal and Tamiš River. It was present at all examined sites in high abundance. In invaded areas, it occurred in lakes and reservoirs [75,76], but a detailed study of water courses in Serbia failed to reveal the presence of the species in the other water bodies in the country denoting that *D. villosus* expand its range only through the rivers closely connected with the main channel of the Danube River.

Another alien species with a broad distribution is *Hypania invalida*. After the initial finding in 1971, it has been identified along the entire stretches of the Danube, Sava, and Tisa river in Serbia [11] at 44 sites. In the following year, another alien species *Branchiura sowerbyi* was discovered. It was observed for the first time in 1972 and since then it could be found in ponds, canals, reservoirs, and lowland rivers in Serbia [6,25,27]. *B. sowerbyi* established its population in the main course of the Danube River and main tributaries Sava, Tisa, and Velika Morava with domination in oligochaetes communities of some localities (e.g., Ljubičevski most). During a detailed investigation of other waterbodies in Serbia in previous years, this species was not observed. Potential negative effects (alterations in macroinvertebrate community composition and effects on the fish community as an alternative host for some fish parasites) on the recipient ecosystem were pointed out in detail [6] but in Serbia, no impacts were observed.

In Serbia, insufficient research has been conducted on invasive alien species and their impacts on the environment. Legislation that regulates the introduction, monitoring, and control of allochthonous

species is the starting point for setting an effective strategy against biological invasions. At the European level, Serbia has ratified the Convention on Biological Diversity and the Berne Convention (Convention on the Conservation of European Wildlife and Natural Habitats), as documents dealing with the issue of invasive species. National legislation concerning this issue is the Law on Nature Protection [77] and the Law on Protection and Sustainable Use of Fish Stock [78]. The mentioned laws cover the subject of invasive species only in the form of general measures and guidelines. Therefore, it is necessary to adopt bylaws that would regulate in more detail the issue of invasive species including the recommendation of measures to control the spread of already introduced taxa as well as to prevent the establishment of newly introduced ones.

In addition to adequate legal frameworks, the list of introduced non-native species and as a further step the assessment of the degree of their invasiveness specified for every species represent the initial, zero state, on the basis of which the effectiveness of control measures could be supervised in the future. In Serbia, there are no national lists of invasive species, with data on their distribution, ecology, and possible impacts on natural ecosystems. So far, preliminary lists of invasive species of plants, fish, amphibians, reptiles, birds, and mammals with a total of 89 taxa have been prepared [79], but these lists do not include invertebrates and fungi. Additionally, it is necessary to have a database of all non-native species of Serbia. It would facilitate data manipulation, update of findings and introduction of new invasive species, assess the current status, and facilitate analysis of various aspects related to invasive species. For macroinvertebrates, such a base already exists, and it is a part of the activities within the Department for Hydroecology and Water Protection at the Institute for Biological Research "Siniša Stanković" [24]. Literary data, information from technical reports, as well as unpublished data of regular monitoring activities available from scientific institutions are included in the database. Research of invasive species is carried out mostly by scientific institutions, but there is a lack of effective cooperation with government agencies. We recommend institutions responsible for surveillance, monitoring, and adequate implementation of legal frameworks to be in close coordination with scientific institutions because the effective strategy against invasive species depends primarily on accurate, updated information and rapid response.

In addition to the taxa list, a fundamental component of risk analysis is the evaluation of invasiveness for every alien species. It is a very complex task for which is necessary to have basic data on the distribution of the species, their abundance, and impact. Based on this assessment species can be assigned to blacklists which may include species that have been identified as highly invasive with severe risk to the environment, economy, or human well-being.

At a national and local level, it is of great importance to involve the public and stakeholders in the process of controlling invasive species. It is necessary to raise the level of awareness about the problem of invasive species through scientific and popular booklets, brochures, leaflets, posters, and tribunes designed for the general public. Additionally, strengthening cooperation with neighboring countries and the sharing of data among countries from all relevant stakeholders is highly recommended.

At the European level, there is no consistency in the incorporation of alien species into water body status assessments according to the WFD. There are few different approaches to classify water bodies [3] e.g., assigning a lower status to a water body based just on the presence of alien species, without taking into account their abundance or impact. A second one is to incorporate registered alien species impact or to use separate biopollution indices for alien species registered in a water body. In order to estimate the level of biological invasions and their impact in Serbia, we used the third approach, separate ecological and risk assessment. As a risk assessment tool, we used the site-specific biocontamination index [80] and depending on both scores general recommendations and measures are being proposed.

The impact of biological invasions on Serbia's biodiversity cannot be accurately assessed, but at the present level of knowledge, it can be concluded that the rate of introduction is increasing. The obtained data can be used for preparing a risk assessment methodology of aquatic invasions and preventing their further expansion and degradation of water habitats.

**Author Contributions:** Conceptualization, K.Z. and M.P.; methodology, M.P., B.T. and K.Z.; validation, K.Z., A.A., B.T., J.T. and M.P.; formal analysis, K.Z. and M.P.; investigation, M.P., B.T., B.V., J.T. and K.Z.; resources, M.P., B.T., K.Z.; data curation, K.Z. and M.P.; writing—original draft preparation, K.Z., A.A., B.V., B.T. and J.T.; writing—review and editing, M.P., K.Z., B.T., A.A., J.T., B.V.; visualization, K.Z., B.V.; supervision, M.P. and B.T.; project administration, M.P.; funding acquisition, M.P. All authors have read and agreed to the published version of the manuscript.

**Funding:** The preparation of the manuscript was supported by the Ministry of Education, Science and Technological Development of the Republic of Serbia, Contract No. 451–03–68/2020–14/200007 and International Commission for the Protection of the Danube River (ICPDR).

**Acknowledgments:** We would like to express our gratitude to our colleagues from the Department for Hydroecology and Water Protection Institute for Biological Research "Siniša Stanković", University of Belgrade and Department for Biology and Ecology, Faculty of Science and Mathematics, University of Kragujevac for their participation in material collection. We would also like to thanks to Goran Poznanović for improvement of the text and constructive comments during preparation of the manuscript.

**Conflicts of Interest:** The authors declare no conflict of interest. The funders had no role in the design of the study; in the collection, analyses, or interpretation of data; in the writing of the manuscript, or in the decision to publish the results.

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
