# Peer review of "Diversity of Alien Macroinvertebrate Species in Serbian Waters"

_water, doi:10.3390/w12123521_

Round 1

Reviewer 1 Report

The resubmission of the paper by Zorić et al., is still very weak and doesn't meet the criteria for publication in Water. Not much improvement has been done from the previous version, the authors need to take time to rework properly on their paper

There is still no clear presentation of the data, a map of distribution of the sampling, distribution of the species, relation between local and invasive species.... statistical analysis. Without all that the paper remain very weak.

Author Response

Response to Reviewer 1 Comments

We are very appreciative for the constructive comments and suggestions that helped us to improve our manuscript. Suggestions and corrections have been accepted and highlighted in the text in Track changes.

General comments: The resubmission of the paper by Zorić et al., is still very weak and doesn't meet the criteria for publication in Water. Not much improvement has been done from the previous version, the authors need to take time to rework properly on their paper.

There is still no clear presentation of the data, a map of distribution of the sampling, distribution of the species, relation between local and invasive species.... statistical analysis. Without all that the paper remain very weak.

Response: The paper is conceived as a presentation of the allochthonous macroinvertebrate species in Serbia, because so far, no paper has been published that would give a comprehensive overview of this type of research in Serbia. As the paper was submitted for a special issue of Water journal, the topic concerning application of WFD after 20 years of adoption, the focus of the manuscript was to link non-native species and the Directive.

We agree that graphic presentation of sampling localities will help to better understand distribution of findings used in this study. To achieve that we provided a map of sampling locations of allochthonous macroinvertebrate species along the rivers in Serbia. Also, in the Appendix 1, a current distribution of alien species in major river basins is presented. If you have a suggestion on how to improve the appearance of the map, we will be pleased to accept it.

Along the entire manuscript mispunctuations, linguistic errors and language style have been corrected and improved as suggested (Lines 26, 47, 48, 52, 59, 68, 71 and also in Table 1).

Point 1: Combine two sentences from the Abstract section (Lines 10-12).

Response 1: Sentences were reformulated in order to avoid duplication of provided information (Lines 10-13): "This article provides the first comprehensive list of alien macroinvertebrate species registered and/or established in aquatic ecosystems in Serbia as a potential threat to native biodiversity."

Point 2: What do you mean candidate county (Line 33)?

Response 2: Serbia is in a process of joining the EU and current status for entry to the European Union is a candidate country. It is official status of the country when the EU Council formally recognises the country as candidate (Line 34).

Point 3: Nothing mentioned (Line 58).

Response 3: Thank you for noticing. Additional information of river system in Serbia has been provided (Lines 63-69) "The river system of Serbia includes ten main drainages, while detailed research has been performed on a total of nine of them – the Danube, Sava, Drina, Kolubara, Zapadna Morava, Južna Morava, Velika Morava, Timok and rivers belonging to the Aegean basin. Although, Tisa river is a tributary of the Danube, it has been presented as separate river system regarding numerous records of allochthonous species."  

Point 4: Which habitats? Too many information missing.

Response 4: This is a description of the Kick and Sweep sampling technique. On the location about 100 m of the watercourse is taken into consideration for sampling including visual assessment of dominant bottom substrate. Multihabitat sampling involves the assessment of all available habitats within a sampling stretch and collection of material from habitats which are represent with more than 5% of total bottom substrate (Line 71).

Point 5: Where this information coming from?

Response 5: The sentence has been changed in order to be clearer (Lines 119-120). In Table 1 along the species name, higher taxa are also stated (family and class).

Reviewer 2 Report

The revised version of the manuscript “Diversity of Alien Macroinvertebrate Species in Serbian Waters” is significantly improved. The authors address more aspects of macroinvertebrate species invasions in Serbian water, including their ways of introduction, their impact on native species and the way they can be incorporated in the WFD.

I have some comments

-I think that a map of the investigated areas (Serbian waters actually) and the locations of the occurrence of the alien species (for all the recorded species) should be added. It would increase the clarity of the observed patterns.

-I believe that the new version should be edited by a native/fluent English speaker, as some added text is missing clarity (I believe due to syntax/grammar errors). For example, in lines 156-158, the sentence doesn’t make any sense to me.

-Lines 144-146 ‘Similar effect may be expected because species was recorded in habitats where finely granulated silty-clay substrate predominates, together with native species: Anodonta anatina, Unio pictorum and Unio tumidus’: I’m not really sure I understand what the authors mean by that. Negative effects on the native species are not recorded/not observed but are expected to occur? And how is the substrate type linked to that?

-Lines 205-208: D. vellosus (I guess D. vellosus should be in line 203 as well?) has been found in the Danube and rivers connected with the Danube that are not reservoirs? I’m not sure I follow the link with the reference. The map would be very helpful to clearly see this pattern

-Lines 216-218: This sentence is too general. What would the observed effects be?

Author Response

Response to Reviewer 2 Comments

Thank you very much for your time and effort to help us improve our work. We have accepted all your suggestions and corrections have been made to the text in Track changes.

Point 1: I think that a map of the investigated areas (Serbian waters actually) and the locations of the occurrence of the alien species (for all the recorded species) should be added. It would increase the clarity of the observed patterns.

Response 1: We agree that graphic presentation of sampling localities will help to better understand distribution of findings used in this study. To achieve that we provided a map of sampling locations of allochthonous macroinvertebrate species along the rivers in Serbia. Accordingly, we modified the text in the Results section, Line 81: "Graphic display of sampling locations was presented in Figure 1." Also, as sites on the map are given as numbers, in Table 1 in front of the species name corresponding numbers are added. Also, in the Appendix 1, a current distribution of alien species in major river basins is presented.

If you have any further suggestion on how to improve the appearance of the map, we will be pleased to accept it.

Point 2: I believe that the new version should be edited by a native/fluent English speaker, as some added text is missing clarity (I believe due to syntax/grammar errors). For example, in lines 156-158, the sentence doesn’t make any sense to me.

Response 2: The sentences have been reformulated and shortened to make them clearer (Lines 191-196). "Although discovered recently, the rapid spread of Dreissena bugensis has been reported. By 2011, the most upstream finding of the species for Serbia was at 1260 rkm at the Ledinci locality [15], while only two years after the species was detected in almost all localities of the middle Danube, with the most upstream locality at 1384 rkm (upstream from the mouth of the Drava River; unpublished data from JDS 3)."

Point 3: Lines 144-146 ‘Similar effect may be expected because species was recorded in habitats where finely granulated silty-clay substrate predominates, together with native species: Anodonta anatina, Unio pictorum and Unio tumidus’: I’m not really sure I understand what the authors mean by that. Negative effects on the native species are not recorded/not observed but are expected to occur? And how is the substrate type linked to that?

Response 3: We intend to point up that possible negative effects may occur due to long-term coexistence of S. woodiana and native Unionidae species, as already been proved in literature, although so far, no such effects in Serbia were observed. In the revised text the sentence has been changed (Lines 184-186): "Similar effect may be expected because species was recorded together with native species: Anodonta anatina, Unio pictorum and Unio tumidus."

Point 4: Lines 205-208: D. villosus (I guess D. villosus should be in line 203 as well?) has been found in the Danube and rivers connected with the Danube that are not reservoirs? I’m not sure I follow the link with the reference. The map would be very helpful to clearly see this pattern.

Response 4: Yes, in Serbia, D. villosus was found only in the main channel of the Danube River, and its tributaries, but not in the reservoirs. The reference we have cited was incorrect, the mistake was corrected and two other relevant references were cited in text (Lines 247-251) and in reference list (Lines 527-536).

Point 5: Lines 216-218: This sentence is too general. What would the observed effects be?

Response 5: The sentence was too general but the possible effects were already given in detailed in the cited reference [5]. As no impacts in macroinvertebrate community were recorded due to the presence of B. sowerbyi on our opinion there was no need for detailed explanation of possible effects. The sentence has been reformulated (Lines 259-262) "Potential negative effects on the recipient ecosystem were pointed out in detail [5] but in Serbia no impacts were observed."

Round 2

Reviewer 1 Report

This new version of the manuscript by Zorić and colleagues is an improved one from my last review. Although they provided a map of the study sites and a better presented list of species. They are still lacking precision on number of organisms observed, number of observation map of distribution of each species, how their number compared to local species, what are there potential threats to local species, what are the environmental controls? what are the different habitats where the species are found. There are still several issues to be dealt with before the paper can be considered publishable.

Author Response

Response to Reviewer 1 Comments

Thank you again for your effort and constructive comments in order to improve the manuscript. Suggestions and corrections have been accepted and highlighted in the text in Track changes.

General comments: This new version of the manuscript by Zorić and colleagues is an improved one from my last review. Although they provided a map of the study sites and a better presented list of species. They are still lacking precision on number of organisms observed, number of observation, map of distribution of each species, how their number compared to local species, what are their potential threats to local species, what are the environmental controls? what are the different habitats where the species are found. There are still several issues to be dealt with before the paper can be considered publishable.

Response:

Along the entire manuscript mispunctuations, linguistic errors and language style have been corrected and improved as suggested (Lines 13, 14, 17, 19, 26, 28, 34, 49, 96, 97, 140, 195, 197, 272, 273, 274, 291, 309, 352, and in Table 1).

After reviewer’s suggestions and comments, the work was elaborated in detail.

In the Materials and Methods section (Lines 60-91) more clearly is described the origin of presented data, from literature source and from field studies conducted by the authors. Also, the description of sampling techniques and methodology was provided in more detail

Result sections was improved with specified distribution of alien species and participation in the community. Also, the most widespread species were pointed out with records during 19 years period.

Discussion was also rearranged and additional information was provided.

Reference list was updated with new literature data and has been cited properly.

Detailed answers, point by point were given below.   

We agree that map of distribution of each species will help to better understand current distribution of the species. Yet, it is very difficult to present every single finding for every species due to large number of records, while only seven species have limited distribution. If you think that it would be more informative to provide such map, we will be pleased to accept your suggestion.

Point 1: According to which references taxonomic identification was performed? Were names checked according to WOMRS? (Lines 72-74).

Response 1: Thank you for noticing. Yes, taxonomic identification was done using WoMRS and reference was added in the reference list.

Point 2: Out of how many autotochnous species? This needs to be properly addressed (Line 76)?

Response 2: Total number of species in Serbia is 995 and this number, with the reference was cited in Lines 184-185.

Point 3: Rivers and borders of the countries confusing. Highlight only the Northern Part as it is where study was conducted (Figure 1).

Response 3: We provided a new map where country borders where properly marked. Also, names of the main rivers of all major river basins were inserted.

Investigations were conducted along the entire territory and we think that map is it is, provide a better insight of a species distribution in the Northern part of the country (Figure 1).

Point 4: Why as appendix as this information is important and should therefore be presented within the main text (Line 111).

Response 4: Thank you for the suggestion. This is an important information, therefore we moved the appendix in the Results section, Lines 176-180.

Point 5: Give numbers not only % (Line 112).

Response 5: The numbers and percentage participation were given for each geographical region (Lines 136-138).

Point 6: How this was tested? No statistical information provided, habitat should be presented before the description of the species. How many times where each species observed? At how many stations? You need as said already several times to go deeper into your analysis. Needs to present clearly first observation and extension range, contribution to the total pool of benthic organisms…. (Lines 116).

Point 7: No information provided in the results regarding this important information (Line 122).

Response 6 and 7: Statistical analyses has not been done, the graph and comments are simply presentation recorded taxa according to the higher classification. In the Results section three paragraphs has been added clearly presenting distribution of the species along the watercourses, percentage participation in the community. Also, the most widespread species where highlighted as well as species with limited distribution and single findings (Lines 149-174).

“Current distribution of the species based on literature data and field investigation in Serbian major river basins is given in the Table 2. Distribution of recorded alien species is predominantly observed along the slow-flowing lowland rivers with a dominance of fine sediment (Danube, Sava, Tisa) and Velika Morava. In total, 28 species were recorded in the Danube, followed by Sava (16 species), Tisa (15 species) and Velika Morava River (10 species). Six macroinvertebrate species were detected in Zapadna Morava basin, while only two alien species were recorded in Južna Morava, Drina and Kolubara basin, and one species in Timok basin. Percentage participation of alien species in the macroinvertebrate community ranged from: 0.9% to 33.33% in the Danube, 1.01% to 17.67% in the Sava, 0.33% to 57.58% in the Tisa and 1.03% to 71.12% in the Velika Morava River. In other watercourses alien species occur irregular in low numbers.

The highest number of alien species per sampling site was recorded in the Danube, locality upstream from the mouth of Velika Morava (14 species). Average number of alien species per river was also the highest in the Danube (8 species) while the average proportion of alien species to a total number of macroinvertebrate taxa was 10.19% in 2007 and much higher in 2013, 25.53%.

The most widespread species in Serbian waters are Physella acuta and Dikerogammarus villosus. P. acuta was recorded at 77 sites in total of eight river basins, predominantly in small and medium sized watercourses up to 500 m above sea level. D. villosus was recorded in 48 sites, along the entire course of the Danube, and localities in the Sava, Tisa, Velika Morava, and canal network in the norther part of the country. Single finding of two specimens was from Ljubišnica (Zapadna Morava basin). Broad distribution within the covered area was assessed also for Branchiura sowerbyi, Sinanodonta woodiana, Corbicula fluminea, Dreissena polymorpha Chelicorophiun curvispinum, and Faxonius limosus. Beside distribution in lowland rivers, findings of D. polymorpha in the Drina and C. curvispinum in the Ljubišnica and Mali Rzav (Zapadna Morava basin) should be point out.

Occasional or single findings of seven species (E. sinensis, Craspedacusta sowerbii, Dugesia tigrina, Hemimysis anomala, Katamysis warpacowsky, Corbicula fluminalis, and Ferrissia fragilis) Serbia indicate that these species have not yet been naturalized in Serbian waters.”

Point 8: Add a table providing info from other countries (Line 123).

Response 8: We consider that very informative for this paper would be lists of alien species in the neighbouring countries. Such overview exists only for Croatia, while in other country in the region there are only single reports (for Bulgaria) and one paper (for Romania). Paragraph was added in the Discussion section (Lines 186-192): “In the neighboring area, the most detailed overview of aquatic alien macroinvertebrate species was given for Croatia [45,46] with complete distribution of species along the waterbodies, possible pathways of introduction and level of biocontamination. No other country in the region has complete data on alien aquatic macroinvertebrate species. In Bulgaria, only three invasive mussels [47] were pointed out as a species of special concern regarding their rapid expansion of distribution range and invasive potential, while in Romania two alien species of EU concern were detected [48].”

Point 9: Not clear enough, you need to provide detail information (Lines 143-144).

Response 9: The sentence has been rephrased and additional information on the distribution of the species has been provided (Lines 211-212).

Point 10: Need to be rephrased (Lines 147-149).

Response 10: The sentence has been omitted from the manuscript as it is not very informative.

Point 11: Which ones (potential negative effects, Line 251)?

Response 11: Thank you for suggestion. Potential negative effects of B. sowerbyi (alterations in macroinvertebrate community composition and effects on fish community as alternative host for some fish parasites) on the recipient ecosystem were added (Lines 341-344).

Point 12: Where is that presented? Defined?

Point 13: Need to be specify for every taxon described.

Response 12 and 13: Our current task is to do detailed assessment of alien species impact on native community and ecosystem using risk assessment methodology and SBC index for biocontamination of sampling site, as a further step toward effective measures against invasive species. Also, application of modified version of another index for invasive species is an ongoing task, and I hope we will have the results soon.

This manuscript is a resubmission of an earlier submission. The following is a list of the peer review reports and author responses from that submission.

Round 1

Reviewer 1 Report

The article presented by Zoric and colleguees on the diversity of alien macroinvertebrate species in Serbian waters is to weak as is for publication. The authors are strongly encourage to work more in depth their dataset and not only present a list of species if they want to be publish in this journal. Other journal such as zootaxa could be considered otherwise. No info on the distribution of the waterways and the species along them are presented. The text should be checked by a native english speaker.

Reviewer 2 Report

The authors in “Diversity of Alien Macroinvertebrate Species in Serbian Waters” provide a list of 29 alien macroinvertebrate species registered in aquatic ecosystems in Serbia, highlighting their Ponto-Caspian origin and their introduction through the Danube. They define their goal as identifying the alien species introduction as a potential threat to native fauna, which would have further implications in the biological quality in the context of the Water Framework Directive. All these are very important, however in the present study the authors fail to catch their goal and this manuscript stays a list of alien macroinvertebrates in Serbia, mostly descriptive on species taxonomy and occurrence, whereas ecological implications are limited to Faxonius limosus. There is definitely a need for such lists, as they can lead further research and management plans; however, I believe that more work has to be done to turn this manuscript to an ecological paper linked to WFD.

For example, most of the alien species mentioned have been recorded for several years. What happened to the native fauna? What are the differences in community structure and biological quality in similar sites where there are no recorded invasions? Is there a need to incorporate e.g. ‘a penalty score’ for invasive species in the metrics used for assessment of biological quality in WFD? How were the species introduced? Natural transport through the Danube? Transportation through ship ballast? This information is limited to 2-3 species in this study.

Specific comments

You mention the year of the first recording of an alien species but what is the current situation? Most of the cited literature is a bit old; up to when is the compiled literature you use for this list?

More than half of the discussion section should be moved to the results section, as it just describes occurrences of species, with no explanation or discussion

Could the fact that most of the alien species are registered in the Danube be a result of more exhaustive research in the area compared to other rivers? You mention JDS a lot. What does it stand for? The Joint Danube Survey? Please specify at first occurrence as not everyone is expected to understand the acronym.